# Acute otitis media symptoms and symptom scales in research with Aboriginal and Torres Strait Islander children

**Penelope Abbott**[1]*, **Caitlin Frede**[1], **Wendy C. Y. Hu**[2], **Sanja Lujic**[3], **Steven Trankle**[1], **Letitia Campbell**[4], **Hasantha Gunasekera**[5], **Robyn Walsh**[1], **Amanda J. Leach**[6], **Peter Morris**[6], **Kelvin Kong**[7], **Jennifer Reath**[1]

1 Department of General Practice, Western Sydney University, Campbelltown, New South Wales, Australia, 2 Medical Education Unit, School of Medicine, Western Sydney University, Campbelltown, New South Wales, Australia, 3 Centre for Big Data Research in Health, University of New South Wales, Sydney, New South Wales, Australia, 4 Kalwun Development Corporation, Gold Coast, Queensland, Australia, 5 Children's Hospital Westmead Clinical School, Sydney University, Sydney, New South Wales, Australia, 6 Child Health Division, Menzies School of Health Research, Darwin, Northern Territory, Australia, 7 School of Medicine and Public Health, University of Newcastle, Newcastle, New South Wales, Australia

* p.abbott@westernsydney.edu.au

**Data Availability Statement:** The interview data obtained throughout this research project cannot be shared publicly as consent was not obtained from participants at the time of recruitment to the

## Abstract

### Background

Aboriginal and Torres Strait Islander children experience a high burden of otitis media. We collected data on symptoms associated with acute otitis media (AOM) in a clinical trial involving children receiving primary care at urban Aboriginal Medical Services. Two scales were employed to monitor symptoms over time: the AOM-Severity of Symptoms scale (AOM-SOS) and the AOM-Faces Scale (AOM-FS). This study took place at a mid-point of the un-blinded trial.

### Methods

We examined symptoms at enrolment and day 7, and compared the scales for trends, and bivariate correlation (Spearman's rho) over 14 days. Responsiveness of the scales to clinical change was determined by Friedman's test of trend in two subgroups stratified by day 7 AOM status. We interviewed parents/carers and research officers regarding their experience of the scales and analysed data thematically.

### Results

Data derived from 224 children (18 months to 16 years; median 3.6 years). Common symptoms associated with AOM at baseline were runny nose (40%), cough (38%) and irritability (36%). More than one third had no or minimal symptoms at baseline according to AOM-SOS (1-2/10) and AOM-FS scores (1-2/7). The scales performed similarly, and were moderately correlated, at all study points. Although scores decreased from day 0 to 14, trends and mean scores were the same whether AOM was persistent or resolved at day 7. Users preferred the simplicity of the AOM-FS but encountered challenges when interpreting it.

study. All other data will be deposited to Western Sydney University's institutional repository (Research Direct), and data access will be controlled by the University's Research Services team, contactable at libraryresearch@westernsydney.edu.au. The author team will be the secondary point of contact for access. The data will be destroyed after 25 years and be unable to be accessed according to the requirements in place at the time of the trials.

**Funding:** Funding for this study was received from the Centre of Research Excellence in Ear and Hearing Health of Aboriginal and Torres Strait Islander Children (http://purl.org/au-research/grants/nhmrc/1078557, authors PA, JR, LC) and the Australian Government National Health and Medical Research Council (https://www.nhmrc.gov.au), grant number GTN1046266, authors JR, PA, HG, AL, KK, PM. The funders did not play any role in the study design, data collection and analysis, decision to publish, or preparation of the manuscript.

**Competing interests:** The authors have declared that no competing interests exist.

## Conclusion

We found minimally symptomatic AOM was common among Aboriginal and Torres Strait Islander children in urban settings. The AOM-SOS and AOM-FS functioned similarly. However, it is likely the scales measured concurrent symptoms related to upper respiratory tract infections, given they did not differentiate children with persistent or resolved AOM based on stringent diagnostic criteria. This appears to limit the research and clinical value of the scales in monitoring AOM treatment among Aboriginal and Torres Strait Islander children.

## Background

Otitis media (OM) is a common disease of childhood characterised by inflammation or infection in the middle ear. Acute OM (AOM) is a middle ear infection most reliably diagnosed by middle ear effusion associated with bulging of the tympanic membrane, and/or ear pain or recent discharge of pus according to Australian guidelines [1]. The natural history of AOM is to resolve spontaneously within 1–2 weeks, although it may persist and lead to chronic OM and conductive hearing loss and other adverse health outcomes [2].

Otitis media occurs more often and at a younger age in Aboriginal and Torres Strait Islander people compared to other population groups in Australia [3]. Aboriginal and Torres Strait Islander people are the First Nations peoples of Australia, and represent multiple distinct linguistic and cultural diverse communities, with the majority living in urban areas [4]. Aboriginal and Torres Strait Islander children are at higher risk for severe chronic OM and long term complications, including persistent suppurative disease and hearing loss [5]. Poor outcomes related to OM are also seen in other Indigenous populations worldwide, and improving ear health in these groups is a recognised research priority [6]. Complicated childhood OM is recognised as a particular burden in remote and semi-remote communities [5], and this is where most OM research has been conducted to date.

AOM symptomatology in Aboriginal and Torres Strait Islander children living in non-remote settings has been little studied. Symptom scales are commonly used in OM research but have not previously been reported in research among Aboriginal and Torres Strait Islander communities. In this paper, we report on symptoms associated with a diagnosis of AOM and two AOM symptom scales used as research tools within 'WATCH', a randomised controlled clinical trial investigating watchful waiting vs. immediate antibiotic therapy for AOM among urban Aboriginal and Torres Strait Islander children [7].

### Symptom scales–The AOM-SOS and AOM-FS

The two scales being used in WATCH are the text-based AOM—Severity of Symptoms Scale (AOM-SOS) [8] and the pictorial AOM-Faces Scale (AOM-FS) [9]. They have been developed and used in North American and European populations. The AOM-SOS was developed as a research tool to track AOM symptoms and assess the effectiveness of different treatments [10]. The AOM-FS was conceived as a clinical tool with research applicability [9].

The questions in the AOM-SOS are based on behaviours which could be observed by carers even in pre-verbal children, and it requires two minutes to complete for literate users under ideal conditions [10]. It has undergone several revisions. Version 3.0 (v3) of the AOM-SOS contained 7 items related to ear tugging, crying, irritability, sleeping, activity, eating, and fever over the last 24 hours, with available responses of 'no, 'a little' or 'a lot' [11]. The corresponding scores of 0, 1 and 2 are summated, giving a total out of 14. Despite ear pain being a common

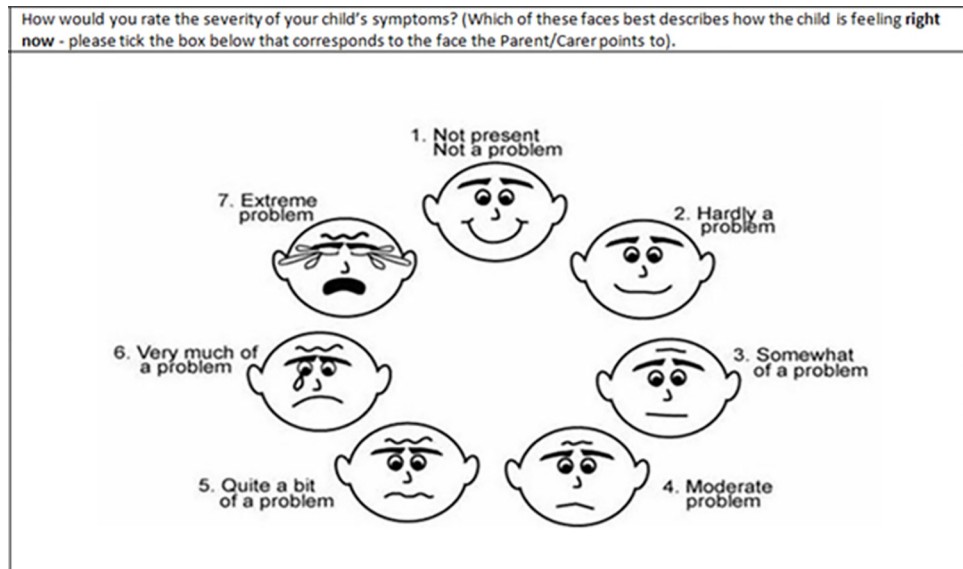

**Fig 1. Acute otitis media–Faces scale.** Faces scale as used in the study and reproduced with permission of the authors [9].

criteria used to differentiate otitis media with effusion from AOM [1, 12], ear pain is not an item on the AOM-SOS as it is not an observable behaviour. The subsequent 5-item AOM-SOS version 4 (v4) had items related to activity and eating removed, bringing the total score to 10 [13, 14]. The minimally important difference using the AOM-SOS(v4) [15] was determined to be a decrease in scores by 55%, representing symptomatic improvement that would be appreciable to carers.

The AOM-FS is a Likert scale, with seven faces in a circular design displaying increasing levels of distress from smiling to crying with verbal descriptors from "not a problem" to "extreme problem", scored 1–7. The carer is requested to circle the face that most applies to their child's wellbeing 'right now' (Fig 1). When combined with otoscopy, the AOM-FS was found to demonstrate excellent sequence validity, concurrent correlation and reliability, but poorer responsiveness when compared to symptom-specific AOM scales which included a question related to pain [9]. The minimally important difference which represents meaningful clinical change has not been reported for the AOM-FS, but research on a 6 item faces scale for pain in children suggested that a drop by 1 face was clinically significant [16].

Our aim in this multiple methods research was to add to knowledge on AOM symptoms in urban Aboriginal and Torres Strait Islander children and to examine the function and usefulness of the AOM-FS and AOM-SOS as symptom severity measures within the WATCH trial.

## Methods

We assessed children's symptoms and the functioning of the two scales using quantitative and qualitative methods. At the time of this research, the WATCH trial was still underway, and participants' data were blinded to study arm.

This research was overseen by six Human Research Ethics Committees (HRECs) with the lead being the Aboriginal Health & Medical Research Council Ethics Committee (AHMRC) (13-2074/13). Written informed consent to participate in this study was obtained from all research officers and the carers of child participants.

## Context: The use of the scales in the WATCH trial

WATCH is a randomised controlled clinical trial of watchful waiting compared to immediate antibiotics for the management of AOM in Aboriginal and Torres Strait Islander children aged between 18 months and 16 years living in urban settings [7]. Eligibility is determined by the presence of middle ear effusion confirmed by tympanometry, and either a bulging tympanic membrane on otoscopy and/ or the presence of ear pain, or irritability in children under the age of 3 years. Children were ineligible if they had conditions prompting immediate antibiotic treatment such as a current or past history of tympanic membrane perforation, craniofacial conditions, or defined systemic features. General practitioners (GPs) diagnosed AOM and determined AOM resolution or persistence at day 7 according to the same criteria. Additional details are provided in Box 1. The AOM (SOS) and AOM- FS were not included in the determination of eligibility for enrolment.

### Box 1. The WATCH trial.

WATCH was a study of whether watchful waiting is non-inferior to immediate antibiotics for AOM in Aboriginal and Torres Strait Islander children attending urban or regional Aboriginal Medical Services for their primary care [7]. The WATCH trial was registered with Australia New Zealand Clinical Trials Registry (ACTRN12613001068752, 24/9/2013). Watchful waiting entails initial observation for persistent symptoms or complications rather than immediate antibiotic treatment and is the recommended treatment in Australia. Although antibiotics may lead to resolution of ear pain 1–2 days earlier, the benefits are considered low compared to side effects and resistance risk. Immediate antibiotics are recommended for Aboriginal and Torres Strait Islander people in remote and semi-remote regions, given their high risk of complications, but there is little evidence in other Aboriginal and Torres Strait Islander communities* [1].

Aboriginal and Torres Strait Islander primary health care services were research partners in the WATCH trial. Research officers (ROs) from the local communities are employed to undertake recruitment and data collection. Children attending the services for any reason underwent ear screening and, if subsequently found to have AOM, participation was invited. GPs made the diagnosis of AOM and decided on eligibility for the study. Parents or carers provided written informed consent for their child to participate. The primary outcome in the trial was clinical resolution of AOM at day 7, determined by GP assessment based on otoscopy, tympanometry and symptoms. The GP answered the question of "Has the Child's index AOM resolved?" choosing between 'Yes/no/don't know'. Time to resolution of AOM symptoms is a secondary outcome.

The scales were employed concurrently in face to face study visits at days 0 and 7 and by phone at days 3 and 14. Trained research officers (ROs) administered the AOM-SOS verbally to parents/carers at each visit. Several other symptom items were incorporated in the study questionnaire, including ear pain and runny nose (Fig 2). We added a response choice of 'don't know' to the AOM-SOS items, intended to allow for the possibility of different carers due to a cultural history of shared caregiving [17].

At face-to-face visits, ROs administered the AOM-FS by showing parents/carers the AOM-FS image and recording responses. At the phone-based study points, ROs read out the

| SYMPTOMS | | | | | | | | | | | |
|---|---|---|---|---|---|---|---|---|---|---|---|
| ASK THESE QUESTIONS OF THE PARENT(S)/CARER. ALL QUESTIONS RELATE TO HOW THE CHILD HAS BEEN FEELING **IN THE LAST 24 HOURS**, COMPARED TO HOW THE CHILD NORMALLY FEELS. | | | | | | | | | | | |
| | | Is this happening? | | | | | Over how many days has this been happening? | | | | |
| | Over the last 24 hours, has the child been… | A little | A lot | No | Don't know | | <1 day | 1 – 2 days | 3 – 6 days | >6 days | Not applicable |
| 6. | Tugging, rubbing or holding the ear(s) more than usual | ☐ | ☐ | ☐ | ☐ | | ☐ | ☐ | ☐ | ☐ | ☐ |
| 7. | Reporting ear pain | ☐ | ☐ | ☐ | ☐ | | ☐ | ☐ | ☐ | ☐ | ☐ |
| 8. | Crying more than usual | ☐ | ☐ | ☐ | ☐ | | ☐ | ☐ | ☐ | ☐ | ☐ |
| 9. | More irritable or fussy than usual | ☐ | ☐ | ☐ | ☐ | | ☐ | ☐ | ☐ | ☐ | ☐ |
| 10. | Having more difficulty sleeping than usual | ☐ | ☐ | ☐ | ☐ | | ☐ | ☐ | ☐ | ☐ | ☐ |
| 11. | Less playful or active than usual | ☐ | ☐ | ☐ | ☐ | | ☐ | ☐ | ☐ | ☐ | ☐ |
| 12. | Eating less than usual | ☐ | ☐ | ☐ | ☐ | | ☐ | ☐ | ☐ | ☐ | ☐ |
| 13. | Having fever or feeling warm to touch | ☐ | ☐ | ☐ | ☐ | | ☐ | ☐ | ☐ | ☐ | ☐ |
| 14. | Vomiting | ☐ | ☐ | ☐ | ☐ | | ☐ | ☐ | ☐ | ☐ | ☐ |
| 15. | Having diarrhoea (≥3 watery stools in 1 day, or ≥2 watery stools for 2 days) | ☐ | ☐ | ☐ | ☐ | | ☐ | ☐ | ☐ | ☐ | ☐ |
| 16. | Coughing more than usual | ☐ | ☐ | ☐ | ☐ | | ☐ | ☐ | ☐ | ☐ | ☐ |
| 17. | Having a runny nose more than usual | ☐ | ☐ | ☐ | ☐ | | ☐ | ☐ | ☐ | ☐ | ☐ |
| 18. | Having a rash | ☐ | ☐ | ☐ | ☐ | | ☐ | ☐ | ☐ | ☐ | ☐ |

**Fig 2. Symptom questions asked of carers of children enrolled in the WATCH trial, inclusive of the AOM SOS scale items (collected at enrolment and days 3, 7 and 14).** *blue shaded items correspond to the Acute Otitis Media-Severity of Symptoms scale (AOM-SOS version 3.0). NB: in AOM-SOS version 4.0, items 11 and 12 above are omitted.

indicators and asked the parents/carers to refer to the image in their take-home materials. Parents/carers were given a take-home diary comprising daily AOM-FS images to collect interim symptoms. They were instructed to circle the image which best corresponded to their child's current symptoms at approximately the same time each day.

## Quantitative methods

We examined the symptoms associated with a diagnosis of AOM at day 0 and 7, namely, the AOM-SOS and AOM-FS scores and individual items, and other reported symptoms collected at the same time. We compared the symptoms reported in children who had persistent or resolved AOM at day 7.

We studied how AOM-SOS and the AOM-FS scales compared in terms of completeness of data and change in scores over time, and whether they yielded similar results as determined by their correlation. Evidence for construct validity can be determined by whether different AOM scales yield similar results [8]. Missing data were tabulated, and changes in scores over time were tested using Friedman's test of linear trend. Nonparametric correlation between the scales at study points between day 0 and14 was measured using Spearman's rho ($r_s$). All tests were two sided, using 0.05 level of significance.

Finally, we studied how responsive the scales were to clinical change between enrolment and day 7 by examining the data on the children who had GP review at day 7. Effectiveness of AOM symptom scales in the research setting can be determined by the degree they detect significant clinical change over time [13]. We divided our sample by whether the children had a

diagnosis of persistent or resolved AOM after their examination at day 7. We then determined the linear trends of AOM-SOS and AOM-FS scores over days 0–14 using the Friedman test of trend. Our hypothesis was that the trends of the scores would differ according to AOM status at day 7.

All data were blinded to trial arm, with data management and analyses carried out using SAS 9.4 software (SAS Institute Inc., Cary, NC, USA) and Python version 3.11.0 (Python Software Foundation). Both version 3 and 4 of AOM-SOS were tested in each analysis to compare their performance. "Don't know" for any item in the AOM-SOS was handled as missing data for the whole scale. "Don't know" responses for the GP adjudication of AOM resolution at day 7 or other symptoms were also handled as missing data. Data from the take-home diary (AOM-FS) were not included in the quantitative analysis.

## Qualitative methods

We used data collected as part of the qualitative arm of the trial [18] to examine the useability of the scales from the perspectives of ROs and parents/carers. This comprised semi-structured interviews face to face or by phone between 2015 and 2020 and a focus group with four ROs on their experience of using the scales and the take-home diary. We extracted data related to the symptom scales into a Microsoft Word document, and generated codes and refined themes in group discussions in an interpretive description analysis approach [19].

## Results

Our dataset included 224 children enrolled in the WATCH trial between 2014 and 2020, ranging in age from 18 months to 16 years (median age 3.6 years; mean age 4.5 years). Males made up 54.5% of the sample. Qualitative data were derived from 11 parent/carer interviews, 12 individual RO interviews and the RO focus group.

## Completeness of scale responses

The highest levels of scale completeness were observed at baseline (AOM-SOS 92%, AOM-FS 99%), and day 7 (AOM-SOS 85%–AOM-FS 88%), with lesser completeness related to missed study visits. The difference between the two scales was due to the option of "don't know" which we had added to the AOM-SOS for our trial and subsequently handled as missing data. There was no significant difference in missing data between version 3 and 4 of the AOM-SOS, p-value = 0.72. The data collected through the take home diary was only available for 55 of 224 participants, and so was not analysed for this paper.

## Correlation and trends of the scores over time

Correlation between the AOM-SOS v 4 and AOM-FS was moderate, increasing over time, from rs = 0.47 at day 0 (p <0.001), 0.55 at day 3 (p<0.001), 0.64 at day 7 (p<0.001) and rs = 0.65 at day 14.

The trends of the scores over time were comparable between the scales (Fig 3). Mean scores for the AOM-SOS version 3 were 5.03 (SD 3.53) on day 0 decreasing to 1.24 (SD 2.53) on day 7 ($\chi^2$(3) = 105.4, p<0,001). For the AOM-SOS version 4 the day 0 score was 3.98 (SD 2.79) decreasing to 1.00 (SD 1.93) by day 7 ($\chi^2$(3) = 102.2, p<0,001). Scores for the AOM-FS were 3.26 (SD 1.75) on day 0 decreasing to 1.75 (SD 1.20) on day 7 ($\chi^2$(3) = 113.8, p<0,001).

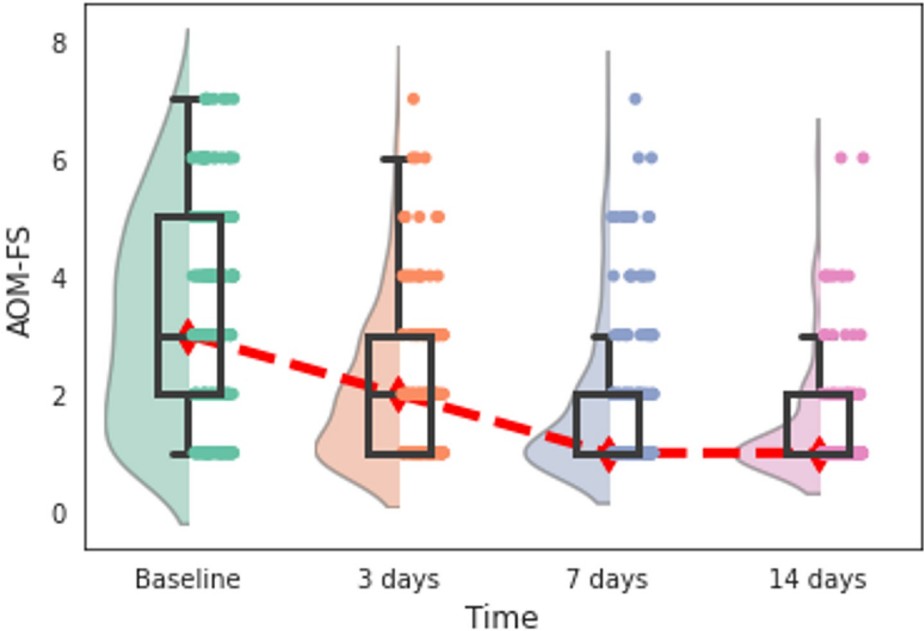

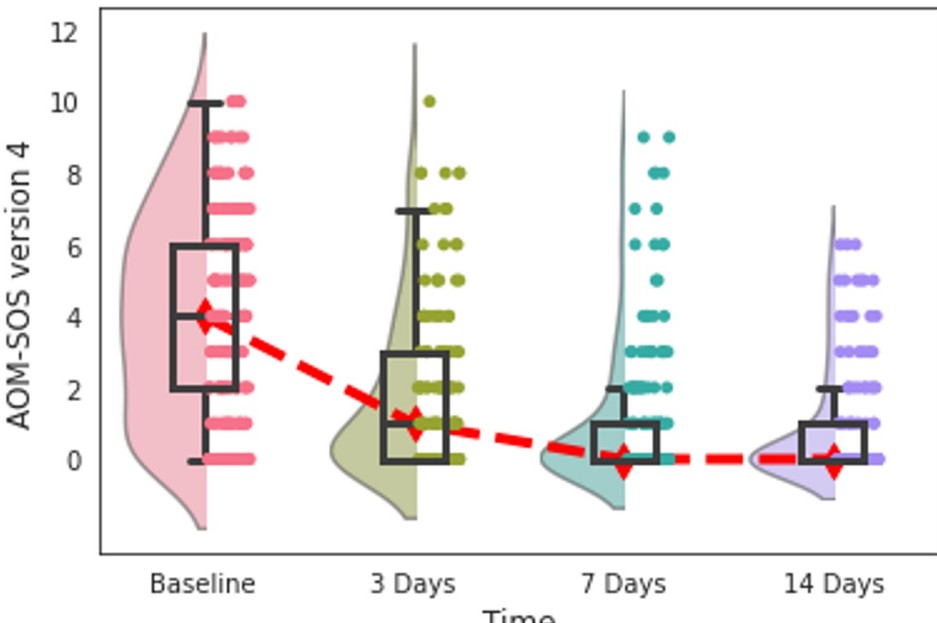

**Fig 3. Distribution of the AOM-FS and AOM-SOS (version 4) symptom scores by days since enrolment into trial.**
The distribution of scores within each time point is plotted using a combination of a violin (represented by a shaded plot), a boxplot (represented by a box), and data points (represented by dots). The width of a violin plot represents the number of results within that score. The boxes indicate the 25th (lower horizontal line), median (middle horizontal line), and 75th (upper horizontal lines) percentiles of the distribution. Dot plots are jittered to increase visibility. The red line indicates the trend in median values over time. Lower values indicate lesser symptom severity. Day 7 and 14 boxplot lines for 25th percentile and median overlap due to more children having lower scores.

## Responsiveness of scales to clinical change

Of the 188 children with day 7 GP assessment data, 113 (60%) had resolved AOM and 75 (40%) had unresolved AOM. Both the AOM-SOS and AOM FS demonstrated responsiveness in that scores decreased in conjunction with expected clinical improvement between day 0 and day 14, with some within-patient variability. However, when the change in scores was stratified by whether AOM was resolved or persistent at the day 7, the decrease in scores occurred regardless of whether AOM had resolved by day 7 or not (Fig 4). There was a small difference in the AOM-FS scores at day 7, with the group with persistent AOM scoring higher than that with resolved AOM, but it didn't reach significance.

## Symptoms at time of diagnosis of AOM at day 0 and day 7

Symptoms documented on recruitment to the study are presented in Table 1. The questions including the symptoms corresponding to items in the AOM-SOS as seen in Fig 2. The most prevalent symptoms, which carers said occurred 'a lot', included runny nose (40%), coughing (38%) and irritability (36%).

Both scales had low scores at baseline in more than a third of children with AOM. A high proportion (72/208 children; 35%) had asymptomatic or minimally symptomatic AOM at day 0 as defined by AOM-SOS v4 scores of 1-2/10. Scores of 3-5/10 were seen in 68 children (35%), scores of 6-7/10 were seen in 42 children (20%) and 26 children (12.5%) had a high symptom burden as determined by a score of 8/10 or greater. This was similar according to the AOM-FS scores at day 0, where 87 (38.8%) children had a score of 1-2/7 indicating 'not a problem' or 'hardly a problem', and 59 (26.3%) children had a score of 5-7/7 on the scale indicating the child had 'quite a bit' to an 'extreme problem'.

The comparison of symptoms associated with resolved AOM and persistent AOM at day 7 is presented in Table 2. Children with AOM resolution had symptom prevalence between 0–10% at day 7, whilst those children with persistent AOM had symptom prevalence of between 1–17%, with coughing and runny nose being the most persistent symptoms in both groups.

## Experiences of using the scales

Themes related to the useability of the scales, including the preference for the AOM-FS over the AOM-SOS and interpretation challenges in the AOM-FS, and the take home diary.

## Use of the scales

The AOM-FS was universally preferred over the AOM-SOS by ROs and parents/carers due to ease of use. Parents/carers preferred describing their child's overall wellness in the AOM-FS over noticing and reporting multiple symptoms for the AOM-SOS. The interactive component of the AOM-FS was particularly valued by parents/carers as it allowed involvement of children in their own health assessment.

*"The kids and them would circle it. . . they'd tell the parents which face they think they are, so it was good and a lot of parents had good feedback." (RO8)*

Most parents/carers and ROs considered the pictorial nature of the AOM-FS to be a strength, particularly for the take-home diary. Most parents/carers found the range and depiction of faces and short descriptions appropriate, but some found interpretation difficult and

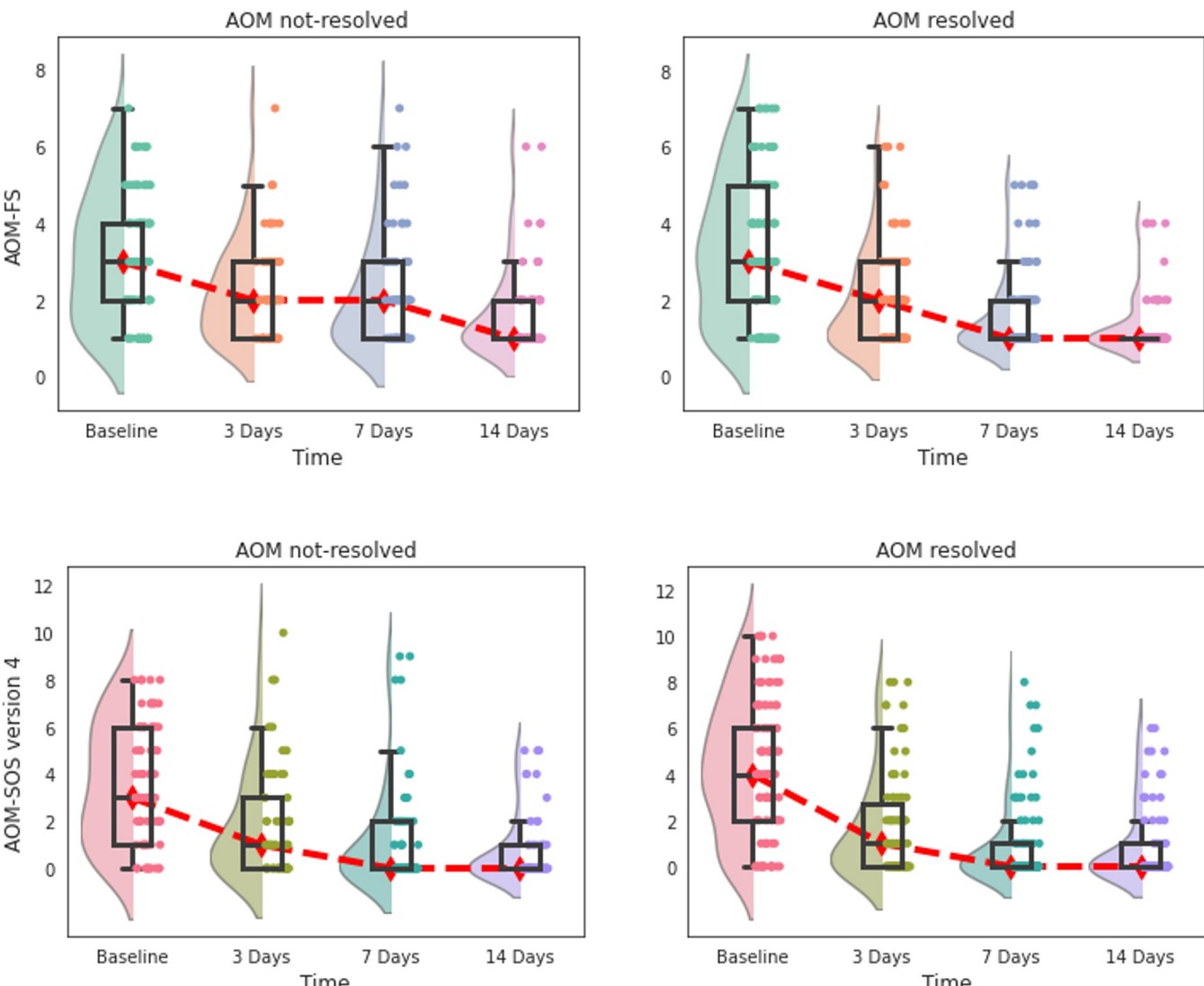

**Fig 4. Responsiveness of AOM-FS and AOM-SOS (version 4) symptom scales to clinical changes by days after enrolment to trial.** The distribution of scores within each time point is plotted using a combination of a violin (represented by a shaded plot), a boxplot (represented by a box), and data points (represented by dots). The width of a violin plot represents the number of results within that score. The boxes indicate the 25th (lower horizontal line), median (middle horizontal line), and 75th (upper horizontal lines) percentiles of the distribution. Dot plots are jittered to increase visibility. The red line indicates the trend in median values over time. Lower values indicate lesser symptom severity. Day 7 and 14 boxplot lines for 25th percentile and median overlap due to more children having lower scores.

would have preferred more verbal or written direction. For example, one carer expressed uncertainty about what to select if the child was not literally crying despite severe pain.

*"I don't think she ever was crying, but there were days when it was really upsetting her, the pain in her ear... So the faces weren't kind of the best I guess of descriptions." (PC2)*

Although the AOM-FS was a global wellness assessment tool, both ROs and parents/carers believed they mainly selected a face on the basis of symptoms they believed were AOM-related, and this could vary between users. Both ROs and parents/carers added written detail at times.

**Table 1. Prevalence of symptoms associated with AOM in children enrolled at baseline (day 0).**

| Symptoms (n = 224) | Prevalence at baseline ('a lot') n (%) | Prevalence at baseline ('a little') n (%) |
|---|---|---|
| Runny nose[a] | 89 (39.7) | 63 (28.1) |
| Coughing[a] | 85 (38.0) | 57 (25.4) |
| Irritable/fussy[b] | 80 (35.7) | 61 (27.2) |
| Difficulty sleeping[b] | 68 (30.4) | 42 (18.8) |
| Ear pain[a] | 62 (27.7) | 47 (21.0) |
| Crying[b] | 58 (25.9) | 58 (25.9) |
| Tugging, rubbing ears[b] | 54 (24.1) | 60 (26.8) |
| Eating less[c] | 40 (17.9) | 58 (25.9) |
| Fever[b] | 34 (15.2) | 69 (40.2) |
| Less playful[c] | 23 (10.3) | 55 (24.6) |
| Vomiting[a] | 5 (2.2) | 17 (7.6) |
| Diarrhoea[a] | 3 (1.3) | 23 (10.3) |
| Rash[a] | 2 (0.9) | 17 (7.6) |

a—Not included in AOM-SOS versions 3 nor 4

b—Included in AOM-SOS version 3 and 4

c—Included in AOM-SOS version 3 only

*"You kind of had to explain it a bit, but I think that's just kids, because she's was like, "I'm happy today," but not realising it's for her ear. . . I mean, she's only six, so I really explained it to her that it's how her ear felt, not how she felt today." (PC6)*

*"You know it's about their ears. The faces are due to their ears, not due to their cold, or something else. If they pick something else up, we write in the page" (RO focus group)*

**Table 2. Severity of symptoms at enrolment into WATCH trial (baseline) compared with primary endpoint time (day 7), stratified by AOM resolution at Day 7.**

| Symptoms ('a lot') | AOM resolution at day 7 (n = 113) | | | Persistent AOM at day 7 (n = 75) | | |
|---|---|---|---|---|---|---|
| | Baseline n (%) | day 7 n (%) | Relative change from baseline (%) | Baseline n (%) | day 7 n (%) | Relative change from baseline (%) |
| Runny nose | 55 (48.7) | 9 (8.0) | -84 | 29 (38.7) | 10 (13.3) | -66 |
| Coughing | 49 (43.4) | 10 (8.8) | -80 | 29 (38.7) | 13 (17.3) | -55 |
| Irritable/fussy | 45 (39.8) | 5 (4.4) | -89 | 26 (34.7) | 8 (10.7) | -69 |
| Difficulty sleeping | 37 (32.7) | 5 (4.4) | -86 | 21 (28.0) | 4 (5.3) | -81 |
| Crying | 33 (29.2) | 7 (6.2) | -79 | 15 (20.0) | 3 (4.0) | -80 |
| Tugging, rubbing ears | 32 (28.3) | 2 (1.8) | -94 | 12 (16.0) | 7 (9.3) | -42 |
| Ear pain | 28 (24.8) | 0 (0.0) | -100 | 21 (28.0) | 8 (10.7) | -62 |
| Fever | 22 (19.5) | 4 (3.5) | -82 | 7 (9.3) | 4 (5.3) | -43 |
| Eating less | 19 (16.8) | 4 (3.5) | -79 | 18 (24.0) | 2 (2.7) | -89 |
| Less playful | 14 (12.4) | 4 (3.5) | -71 | 6 (8.0) | 1 (1.3) | -83 |
| Vomiting | 3 (2.7) | 1 (0.9) | -67 | 1 (1.3) | 1 (1.3) | 0 |
| Diarrhoea | 2 (1.8) | 1 (0.9) | -50 | 0 (0.0) | 1 (1.3) | N/A |
| Rash | 1 (0.9) | 1 (0.9) | 0 | 1 (1.3) | 2 (2.7) | 100 |

N/A = not applicable

*A major limitation of the AOM-FS according to parents/carers and ROs was it did not address variability of symptoms over the day given it was a 'right now' scale. Parents/carers perceived this could inaccurately represent how their child had been feeling, including due to the effect of analgesia. Several reported choosing the most representative face for the previous 24 hours instead of at the time they filled out the scale.*

"Through the day they were okay, afternoon they played up a bit, night time they played up. So, if I did the diary entries in the morning, so, I miss that afternoon or that night time one where I think, isn't it more important to know when they're playing up with their ears?" (PC3)

### Views on the take home diary

Although parents/carers reported the diary was easy to fill out, few were completed and the diary was considered impractical by parents/carers and ROs. Several parents/carers declined to take it home. Particular barriers were competing priorities, and the tedium of completing it when symptoms had resolved.

*"I couldn't keep remembering to do it. . . having to remember to give her medication how many times a day plus having other children isn't ideal, or isn't realistic that people are going to be able to keep up with it." (PC2)*

Both parents/carers and ROs expressed concerns about the diary's accuracy. Some children had multiple carers or attended childcare and so different people completed the diary, often at varying times. ROs noted that some parents/carers did not fill out the diary every day as instructed but appeared to have done it when they returned it at day 7.

*"They would tell you to circle same time every day–so you know it's hard if parents/carers are at work and the child is at day-care—they just filled it out whenever" (RO focus group)*

### Discussion

We examined symptoms associated with AOM and compared the performance of two AOM symptom scales during a community based RCT with Aboriginal and Torres Strait Islander children living in urban settings in Australia. Neither scale had been previously used as research tools in this population. In planning the RCT, we thought the simplicity of the pictorial AOM-FS may assist in data collection yet provide useful symptom data, as compared to the more commonly used AOM-SOS. We found that the AOM-FS, although not primarily designed as a research scale, was comparable to the AOM-SOS in terms of completeness of data, correlation and linear trends. However, our findings challenge the usefulness of both the scales in this and other otitis media research with Aboriginal and Torres Strait Islander children. A substantial number of children had no or minimal symptoms at the time they were diagnosed with AOM, and there was a trend of decreasing symptoms over 2 weeks, regardless of whether children's AOM had resolved by day 7 or not according to stringent AOM diagnostic criteria. This could be due to the scores in the AOM-SOS and AOM-FS at baseline both reflecting upper respiratory tract infections rather than AOM symptoms.

Research and clinical management of AOM is complicated by the fact that different diagnostic criteria for AOM may be applied internationally [20–22]. The criteria for an AOM diagnosis which determined recruitment to the WATCH trial was bulging of the tympanic membrane and/or ear pain (or irritability in children under 3 years). This accords with

Australian guidelines and is congruent with the trend to de-emphasise symptoms in the diagnosis of AOM [1, 12, 21]. However debate continues and it is common for guidelines and research protocols to include recent onset of symptoms in the diagnosis of AOM [20–22]. For example, in a US-based clinical trial of antibiotics versus placebo for the management of AOM in children under the age of 2 years, children with minimal symptoms as determined by a AOM-SOSv3 score of 0-2/14 were not eligible for the study [23].

More than a third of the children with AOM in the WATCH trial had no or minimal symptoms at the time of recruitment as scored by both the AOM-SOS and AOM-FS scales. It is known that otitis media with effusion is commonly asymptomatic as per parental report [24], however asymptomatic or minimally symptomatic AOM has been less studied. Asymptomatic AOM, diagnosed by a bulging tympanic membrane, was common in Aboriginal and Torres Strait Islander children living in remote communities participating in ear screening [25]. Our research suggests a large number of children living in urban areas may also have minimally symptomatic AOM. Our findings support the need for routine ear screening of Aboriginal and Torres Strait Islander children who are at risk of AOM given that AOM may not be detected if only symptomatic children are examined.

It is known that AOM is usually associated with viral upper respiratory tract infections [12, 26]. In our sample a cough and runny nose were the most common reported symptoms at baseline, neither of these items being included in the AOM-SOS scale. The difficulty in distinguishing between symptoms related to upper respiratory tract infections and AOM has been raised. A Finnish study found that parental suspicion of AOM and parental symptom report, including through use of the AOM-SOS and the AOM-FS, did not predict AOM in children aged 6 to 36 months [27]. The researchers subsequently tested adapted pain scales in similarly aged children with respiratory tract infections and suspected AOM to determine if they could assist carers to identify if their child had AOM, but these were also unsuccessful at differentiating AOM from respiratory tract infections [28]. In a study of children younger than 12 months with upper respiratory tract infections, higher scores of parental-reported symptoms (fever, earache, poor feeding, restless sleep, and irritability) were statistically associated with the prediction of AOM. However, these symptoms were most predictive of AOM when combined with reports of cough, severe ear pain, and the AOM risk factor of day care attendance [21].

Our participants, at a median age of 3.5 years, were older than most of the children involved in testing and research use of the AOM-SOS and AOM-FS, who were largely under 24 months [9, 10, 15, 27]. This could affect the functioning of the scales, including because older children are better able to communicate their symptoms, decreasing reliance on parental report of behaviours. However, this adds strength to our findings that a relatively large proportion of children had asymptomatic or minimally symptomatic AOM.

One-item symptom scales like the AOM-FS have been little used in research and it can be difficult to know what such scales are measuring. Faces scales are preferred by respondents when given a choice in assessing pain in children [29]. The visual AOM-FS was strongly preferred in our research too, though was not a successful research tool in the paper-based home diary form. Our qualitative findings suggest a value in the AOM-FS was allowing inclusion of children in scoring their symptoms. However, user focus on symptoms they perceived to be ear-related instead of global wellness may have led to inconsistency or underrepresentation of symptoms possibly related to AOM or its treatment but not recognised as such. Another disadvantage was its function as a 'right now' tool which parents/carers felt did not provide a true representation of their child's health.

The study has some limitations. The criteria we used for the diagnosis of AOM, namely tympanic membrane bulge and/or ear pain, determined our participant group and our study must be interpreted accordingly. Furthermore, it is possible that GPs did not recruit highly

symptomatic children into the WATCH trial, preferring to prescribe antibiotics outside guideline recommendations. The relatively large proportion of children in our study who had limited symptoms at baseline affected how the scales functioned. We did not have enough participants with high symptom scores to be able to assess whether the symptom scales are more useful in that subpopulation. Other potential limitations are the correlation of two scales with differently defined time capture and the administration at some study points by phone, which is likely to have changed their functioning. This particularly applied to the AOM-FS, given that it is likely carers did not always refer to the faces image in scoring their child's symptoms over the phone.

We note that the AOM-SOS was not designed to discriminate between those with AOM and those without, but to follow the course of the disease according to score changes compared by treatment groups at designated time points [8, 10]. Our study compared changing scores, whereas it has been recommended that it may be more useful to report percentage of change in participant scoring rather than actual scores when researching different AOM treatments [15]. However, this would not have ameliorated the fundamental problem that scores were low at baseline for a large proportion of participants and did not differ according to AOM status at day 7. Finally, given the trial is ongoing and our data remains blinded, we have not compared scores between treatment groups. Differentiating potential effects individually of these two groups in the future may add understanding of the functioning of the scales.

## Conclusion

Our research provides new information on AOM symptoms in urban Aboriginal and Torres Strait Islander children. Minimally symptomatic AOM appears to be common in Aboriginal and Torres Strait Islander children living in urban communities. Given their high burden of ear and hearing problems, routine and opportunistic ear screening may be necessary to adequately detect AOM.

Our examination of the function and usefulness of the AOM-FS and AOM-SOS scales as research tools in the WATCH trial leads us to question the value of these scales in our study population. Despite functioning similarly, the AOM-SOS and AOM-FS symptom scales did not discriminate between children with resolved and persistent AOM and it appears likely the scales were largely measuring symptoms related to concurrent upper respiratory tract infection. The value of the scales as research or clinical tools among Aboriginal and Torres Strait Islander children, a population at high risk of otitis media and its complications, appears limited.

## Supporting information

**S1 File. Global health questionnaire.**
(DOCX)

## Acknowledgments

We acknowledge our AMS partners in this research, both the health services and the communities they serve, including the Institute for Urban Indigenous Health, Kalwun Development Corporation, Southern Queensland Centre of Excellence in Aboriginal and Torres Strait Islander Primary Health Care, Tharawal Aboriginal Corporation, Townsville Aboriginal and Islanders Health Services, Winnunga Nimmityjah Aboriginal Health Service and Yerin Eleanor Duncan Aboriginal Health Centre. We also acknowledge the broad research team of

WATCH and INFLATE. Thank you to Dr Nicholas I-Hsien Kuo for producing the figures and Ms Zhisheng Sa for assistance with statistical analysis.

## Author Contributions

**Conceptualization:** Penelope Abbott, Wendy C. Y. Hu, Sanja Lujic, Steven Trankle, Hasantha Gunasekera, Amanda J. Leach, Peter Morris, Kelvin Kong, Jennifer Reath.

**Data curation:** Penelope Abbott, Caitlin Frede, Sanja Lujic, Steven Trankle, Letitia Campbell, Robyn Walsh.

**Formal analysis:** Penelope Abbott, Caitlin Frede, Wendy C. Y. Hu, Sanja Lujic, Steven Trankle, Letitia Campbell, Hasantha Gunasekera, Amanda J. Leach, Peter Morris, Jennifer Reath.

**Funding acquisition:** Penelope Abbott, Wendy C. Y. Hu, Hasantha Gunasekera, Amanda J. Leach, Peter Morris, Kelvin Kong, Jennifer Reath.

**Methodology:** Penelope Abbott, Wendy C. Y. Hu, Sanja Lujic, Steven Trankle, Hasantha Gunasekera, Robyn Walsh, Amanda J. Leach, Peter Morris, Kelvin Kong, Jennifer Reath.

**Project administration:** Penelope Abbott, Robyn Walsh.

**Writing – original draft:** Penelope Abbott.

**Writing – review & editing:** Caitlin Frede, Wendy C. Y. Hu, Sanja Lujic, Steven Trankle, Letitia Campbell, Hasantha Gunasekera, Robyn Walsh, Amanda J. Leach, Peter Morris, Kelvin Kong, Jennifer Reath.

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
