## [Decision Letter · Decision Letter 0]

2 Oct 2022

PONE-D-22-18129Acute otitis media symptoms and symptom scales in research with Aboriginal and Torres Strait Islander childrenPLOS ONE

Dear Dr. Abbott,

Thank you for submitting your manuscript to PLOS ONE. After careful consideration, we feel that it has merit but does not fully meet PLOS ONE’s publication criteria as it currently stands. Therefore, we invite you to submit a revised version of the manuscript that addresses the points raised during the review process.

We look forward to receiving your revised manuscript.

Kind regards,

Shengwen Calvin Li, PhD

Academic Editor

PLOS ONE

Journal Requirements:

2. Please include a complete copy of PLOS’ questionnaire on inclusivity in global research in your revised manuscript. Our policy for research in this area aims to improve transparency in the reporting of research performed outside of researchers’ own country or community. The policy applies to researchers who have travelled to a different country to conduct research, research with Indigenous populations or their lands, and research on cultural artefacts. The questionnaire can also be requested at the journal’s discretion for any other submissions, even if these conditions are not met.  Please find more information on the policy and a link to download a blank copy of the questionnaire here: https://journals.plos.org/plosone/s/best-practices-in-research-reporting. Please upload a completed version of your questionnaire as Supporting Information when you resubmit your manuscript

Additional Editor Comments (if provided):

Academic Editor's Notes:

1) The current version is preliminary. After careful consideration, we feel that it has merit but is not suitable for publication as it currently stands. We uphold the standard and integrity of [PLOS ONE] with rigorously constructive critiques, which indicated in an impact factor of 3.752, substantially higher than the average Journal of IF 1.0 out of 29,000 Journals collected by The Web of Science™ platform (Clarivate) that is the world's most trusted publisher-independent global citation database. Below are specific comments and suggestions for the authors that should be incorporated to improve its clarity, coherence, and logic flow.

2) All the Figures (1-4): Figure titles must carry self-explanatory information. An ideal figure title should give complete information to the reader even without reading the text. All the figure legend descriptions are not written in keeping this point in mind. All of these should be placed under the Figures; neither split part of the above and part of the below (e.g., Fig 3, Fig 4), nor all above the Figures.

3) Lines 149 – 154: Brief descriptions should be provided in the figure legend, as resonated with the principle that an ideal figure legend should give complete information to the reader even without reading the text. E.g., * shaded items correspond to the AOM-SOS version 3.0 – what was the version?

4) Besides what was presented in the manuscript, are there other demographic characteristics (e.g., laboratory results that might contribute to the syndromes and imaging upper respiratory tract infections)? A few specific cases (individual patients) will be helpful to illustrate the complexity of the disease. Some outcome measurements might be relevant.

5) Lines 111 – 113: "Figure 1 Acute Otitis Media – Faces Scale" the circular scheme of the current version is confusing. One dimension of the schematic diagram might better illustrate escalated scaling, which should be more concrete in the frequent description.

6) Lines 411-420: A schematic diagram showing the different criteria for an AOM diagnosis between the international standard and the method used in the manuscript should be provided to govern the current study. Ideally, a measurement matrix derived from the study should be proposed, i.e., a diagnostic flow chart for both AOM-SOS and AOM-FS symptom scales.

7) Conclusions: Write brief statements on the results obtained per the study's objectives.

Reviewers' comments:

Reviewer's Responses to Questions

**Comments to the Author**

1. Is the manuscript technically sound, and do the data support the conclusions?

Reviewer #1: Yes

Reviewer #2: Yes

2. Has the statistical analysis been performed appropriately and rigorously? 

Reviewer #1: No

Reviewer #2: Yes

3. Have the authors made all data underlying the findings in their manuscript fully available?

Reviewer #1: No

Reviewer #2: No

4. Is the manuscript presented in an intelligible fashion and written in standard English?

Reviewer #1: Yes

Reviewer #2: Yes

5. Review Comments to the Author

Reviewer #1: In this paper, the authors studied otitis media. They compared the scales for trends, and bivariate correlation (Spearman’s rho) over 14 days. Responsiveness of the scales to clinical change was determined by Friedman’s test of trend in two subgroups stratified by day 7 AOM status.

I have the following questions.

1. No real figure legend is provided. “Note” is added at the end of Figures 3 and 4. But a real figure legend is needed.

2. In Figure 4, some days show weird string-like shape. It seems that the values are either discrete integers, or close to integers. Why is this? Any such trend is not shown in other days? Full explanation is needed.

3. Figure 4. Better visualization is needed. I do not think violin plot is the good choice here. Better label is also needed here.

4. Table 2. Second line from last, right column, the authors claim that the relative change from base line is 100%. But this is incorrect. The relative change is in fact infinity (or does not exist) since it is equal to 1/0.

5. Page 12, line 218. What is “rs”. This is not standard notation. I suspect that authors want to use Pearson correlation “r” but I am not sure. Please double check and provide explanation. What are the p-values for day 3 and day 7?

Reviewer #2: The manuscript presents a detailed and rigorous analysis of the use of two different scales – AOM-SOS and AOM-FS – to monitor AOM in diagnosed patients. The strength of the manuscript is the critical analysis of the limitations of the study, namely, low symptom severity at enrollment, difficulty distinguishing between symptoms related to AOM and upper respiratory tract infections, as well as the limitations of each scale system. I only have a few minor suggestions and requests:

1. Label figure panels with letters and refer to them accordingly in the captions and the text.

2. State the statistical significance of the decrease in AOM-SOS and AOM-FS scores over time (Figs. 3-4 and corresponding sections in the main text).

3. I recommend splitting the second sentence of the abstract into two sentences: “We collected data on symptoms associated with acute otitis media (AOM) in a clinical trial involving children receiving primary care at urban Aboriginal Medical Services. Two scales were employed to monitor symptoms over time: the AOM-Severity of Symptoms scale (AOM-SOS) and the AOM-Faces Scale (AOM-FS)”.

6. PLOS authors have the option to publish the peer review history of their article (what does this mean?). If published, this will include your full peer review and any attached files.

Reviewer #1: No

Reviewer #2: No

---

## [Author Response · Author response to Decision Letter 0]

17 Dec 2022

Dear team

Please find attached document with our detailed response to the editor and reviewers.

regards

Penelope Abbott

---

## [Editor Report · Decision Letter 1]

12 Jan 2023

Acute otitis media symptoms and symptom scales in research with Aboriginal and Torres Strait Islander children

PONE-D-22-18129R1

Dear Dr. Abbott,

We’re pleased to inform you that your manuscript has been judged scientifically suitable for publication and will be formally accepted for publication once it meets all outstanding technical requirements.

Kind regards,

Shengwen Calvin Li, PhD

Academic Editor

PLOS ONE

Additional Editor Comments (optional):

R1 can be accepted as the authors have fully addressed the rigorous comments in two peer-review reports given the lack of clinical trials on children has been historically evident.
---

## [Editor Report · Acceptance letter]

14 Feb 2023

PONE-D-22-18129R1 

Acute otitis media symptoms and symptom scales in research with Aboriginal and Torres Strait Islander children 

Dear Dr. Abbott:

I'm pleased to inform you that your manuscript has been deemed suitable for publication in PLOS ONE. Congratulations! Your manuscript is now with our production department. 

Kind regards, 

on behalf of

Prof. Shengwen Calvin Li 

Academic Editor

PLOS ONE